# The Potential Mechanism of Bufadienolide-Like Chemicals on Breast Cancer via Bioinformatics Analysis

**DOI:** 10.3390/cancers11010091

**Published:** 2019-01-14

**Authors:** Yingbo Zhang, Xiaomin Tang, Yuxin Pang, Luqi Huang, Dan Wang, Chao Yuan, Xuan Hu, Liping Qu

**Affiliations:** 1Tropical Crops Genetic Resources Institute, Chinese Academy of Tropical Agricultural Sciences, Danzhou 571737, China; zhangyingbo1984@catas.cn or zhangyingbo1984@163.com (Y.Z.); wang_dan1414@163.com (D.W.); yuanchao79@126.com (C.Y.); mchuxuan@163.com (X.H.); 2Hainan Provincial Engineering Research Center for Blumea Balsamifera, Danzhou 571737, China; 3School of Traditional Chinese Medicine, Guangdong Pharmaceutical University, Guangzhou 510006, China; txm1209@163.com; 4National Resource Center for Chinese Materia Medica, China Academy of Chinese Medical Sciences, Beijing 100700, China; 5College of Pharmacy, Chengdu University of Traditional Chinese Medicine, Chengdu 611137, China; quliping2018@163.com

**Keywords:** Bufadienolide-like chemicals, molecular mechanism, anti-cancer, bioinformatics

## Abstract

Bufadienolide-like chemicals are mostly composed of the active ingredient of Chansu and they have anti-inflammatory, tumor-suppressing, and anti-pain activities; however, their mechanism is unclear. This work used bioinformatics analysis to study this mechanism via gene expression profiles of bufadienolide-like chemicals: (1) Differentially expressed gene identification combined with gene set variation analysis, (2) similar small -molecule detection, (3) tissue-specific co-expression network construction, (4) differentially regulated sub-networks related to breast cancer phenome, (5) differentially regulated sub-networks with potential cardiotoxicity, and (6) hub gene selection and their relation to survival probability. The results indicated that bufadienolide-like chemicals usually had the same target as valproic acid and estradiol, etc. They could disturb the pathways in RNA splicing, the apoptotic process, cell migration, extracellular matrix organization, adherens junction organization, synaptic transmission, Wnt signaling, AK-STAT signaling, BMP signaling pathway, and protein folding. We also investigated the potential cardiotoxicity and found a dysregulated subnetwork related to membrane depolarization during action potential, retinoic acid receptor binding, GABA receptor binding, positive regulation of nuclear division, negative regulation of viral genome replication, and negative regulation of the viral life cycle. These may play important roles in the cardiotoxicity of bufadienolide-like chemicals. The results may highlight the potential anticancer mechanism and cardiotoxicity of Chansu, and could also explain the ability of bufadienolide-like chemicals to be used as hormones and anticancer and vasoprotectives agents.

## 1. Introduction

Despite considerable efforts on early diagnosis and treatment in the last decade, breast cancer remains the most common malignancies for women worldwide, representing ~22% of female malignancies [1,2,3,4]. In addition to early diagnosis, new chemotherapeutic agents and more effective therapies are needed to reduce mortality. Traditional Chinese medicine has existed for thousands of years and can treat cancer. Chansu is one of the most famous traditional Chinese medicines. It has been used for centuries in various aspects, such as anaesthesia for anesthesia, antitumor, anti-inflammation, and anti-arrhythmia conditions [5,6,7,8]. Chansu is mostly from the glandular secretion and dried product of *Bufo* bufo gargarizans Cantor or *Bufo* melanostictus Schneider [8]. It includes resibufogenin, bufalin, arenobufagin, cinobufagin, bufotoxin, telocinobufagin, bufotaline, and cinobufotalin [5,6,8] (Figure 1).

Over the last decade, many groups have studied the pharmacological activities and antitumor activity of bufadienolide-like chemicals. For example, Li et al. [9] reported that cinobufagin has significant cancer-killing capacity for a range of cancers, including HCT116 cells, HT29 cells, A431 cells, PC3 cells, A549 cells, and MCF-7 cells. Mechanistic studies showed that cinobufagin can induce tumor cells apoptosis and modulate hypoxia-inducing factor-1 alpha subunit (HIF-1α). Yeh et al. [10] and Yu et al. [11] reported that bufalin and cinobufagin have a potent inhibiting effect on androgen-dependent and -independent prostate cancer cells, similar to Dong et al. [12], Wang et al. [13], and Ko et al. [14] via HepG2 cells, T24 cells, and HeLa cells.

Immunotherapy, an evolving approach for the management of triple negative breast cancer: Converting non-responders to responders. These results demonstrate that Chansu is a potent anticancer agent for a range of cancers, but its potential anticancer mechanisms are unclear. Here, the gene set variation analysis (GSVA) algorithm [15] was first used to identify differentially expressed genes (DEGs) and relative enrichment pathways underlying eight bufadienolide-like chemicals. A series of bioinformatics analyses, including gene enrichment analysis, tissue-specific co-expression network construction, and differentially-regulated sub-network detection, can relate the findings to the breast cancer phenome and hub gene selection. The relation to survival probability and similar small-molecule detection used the DEGs with relative enrichment pathways (Figure 2). This work shows the potential mechanism of bufadienolide-like chemicals on breast cancer, especially differentially regulated sub-networks that relate to breast cancer and hub genes disturbed by bufadienolide-like chemicals. This work highlights the potential application of bufadienolide-like chemicals on breast cancer, especially as a novel agent for cancer therapy.

## 2. Results

### 2.1. Identification of DEGs

Based on the differentially expressed genes analysis associated with the gene sets enrichment variation analysis strategy, a total of 80 differentially expressed genes (DEGs) involved in the 44 MSigDB C2 curated gene sets were identified (Figure 3A,B), and the top 20 DEGs’ expression heatmap is shown in Figure 3C. Of which, 38 genes involved in the Singh NFE2L2 targets gene sets, Chang dominant negative gene sets immortalized by HPV31 and Lin silenced gene sets by tumor microenvironment were up-regulated (Appendix A), including IF16 (interferon-inducible protein 6), IRF9 (interferon regulatory factor 9), IFIT1 (IFN-induced protein 1 with tetratricopeptide repeats), ISG15 (Interferon-stimulated gene 15), BST2 (bone marrow stromal cell antigen 2), OAS3 (2′-5′-oligoadenylate synthetase 3), OAS1 (2′-5′-oligoadenylate synthetase 1), DDX60 (DEAD box polypeptide 60), CYP1A1 (cytochrome P450 1A1), CEACAM6 (carcinoembryonic antigen-related cell adhesion molecule 6), keratin genes KRT81, and so on.

Among the differentially expressed genes associated with enrichment gene sets, 42 genes involved in the 41 gene sets were down-regulated (Appendix A), such as the genes involved in the Iizuka (Appendix A) liver cancer progression pathway, including PPIF (peptidylprolyl isomerase F), TMED2 (transmembrane trafficking protein 2 with emp24 domain), SAFB (scaffold attachment factor B), SQLE (squalene epoxidase), PICALM (phosphatidylinositol binding clathrin assembly protein), STIP1 (stress-induced phosphoprotein 1), CYB561 (cytochromes b561), CCT2 (chaperonin 2β with TCP1 domain); the genes sets involved Thum systolic heart failure pathway, including CCNG2 (cyclin G2), TMED2 (transmembrane emp24 domain trafficking protein 2), FH (fumarate hydratase), TAF9B (ATA-box binding protein associated factor 9b), CCT2 (chaperonin-containing t-complex polypeptide 1 beta), transmembrane receptor NOTCH2, PICALM (subfamily A (MS4A), and CCNL2 (cyclin L2); and also Reactome DNA strand elongation, Reactome regulated proteolysis of P75NTR, and other gene sets were downregulated, with logFC form −0.89~−0.27.

In order to obtain a biological interpretation of those genes in the GO and KEGG pathway functional groups, GO and KEGG enrichment analysis were performed with the clueGO plug [23] in Cystoscape [24]. Results indicated that those genes that were up-regulated were rich in terms of type I interferon signaling response to virus, defense to other organisms, regulation of viral genome replication, and 2′-5′-oligoadenylate synthetase activity, and those activated may be because of the up-regulation of IRF9, IFI6, IFI27, ISG15, IFIT1, OAS1, and OAS3 (Figure 4A). Further investigation with the KEGG pathway enrichment analysis showed those up-regulated genes could cause the activation of the IFN-induced pathway, type II interferon signaling pathway, and regulation of protein ISGylation by the ISG15 deconjugating enzyme USP18 pathway (Figure 4B). The genes that were down-regulated were rich in terms of protein kinase complex, transcription factor TFTC complex-1, the SAGA-complex, and cargo loading into vesicle (Figure 4A). Further investigation with KEGG pathway enrichment analysis showed those genes could negatively affect the transport of fringe-modified NOTCH to the plasma membrane pathway (Figure 4B).

### 2.2. Similar Small Molecule Detection

Detection of the similar small molecule with the Comparative Toxicogenomics Database (CTD) (http://ctdbase.org/) [16] and connectivity map (CMAP2) (https://portals.broadinstitute.org/cmap/) [17,18] database provides a better understanding the molecular mechanism of bufadienolide-like chemicals, and its potential value as a novel agent for cancer therapy. Based on the results with detecting the CTD Database, valproic acid, cyclosporine, and estradiol had the most similar target with bufadienolide-like chemicals (Figure 5). Valproic acid, a histone deacetylase inhibitor, which once was widely used as an antiepileptic, has recently also shown anti-cancer activity in an vitro/vivo model [25]. Estradiol is a sex hormone with anticancer activity, and is also widely used for the treatment of breast cancer, especially for postmenopausal women [26,27,28].

Based on the results from the CMAP2 database (https://portals.broadinstitute.org/cmap/) [17,18], V03AF, G03GB, C05AX, and C05CX were the top matching drugs with bufadienolide-like chemicals (Table 1). V03AF, a type of detoxifying agent for antineoplastic treatment, had an opposing effect on the expression of bufadienolide-like chemicals. This result provided evidence for bufadienolide-like chemicals’ potential value as a novel agent for cancer therapy. G03GB, one type of sex hormone and a modulator of the genial system, had the most similar expression profile with bufadienolide-like chemicals. This means the bufadienolide-like chemicals also use estradiol, epimestrol and cyclofenil in breast cancer. C05AX and C05CX are two types of vasoprotectives agents, indicating that bufadienolide-like chemicals also have a potential use as vasoprotectives-like drugs.

From the evidence from detecting the similar small molecules with the CTD database and CMAP2 database, it was indicated that bufadienolide-like chemicals were one kind of steroid with the same physiological activity as estradiol and G03GB (ATC code), with potential value for use in cancer, especially breast cancer.

### 2.3. The Tissue Specific Co-Expression Network and Breast Cancer Associated Subnetwork Regulated by Bufadienolide-Like Chemicals

It is clear that most of the genes exert their function by collaborating with other genes in the network, which represent rigid molecular machines, cellular structures, or dynamic signaling pathways [29]. Here, a breast tissue specific co-expression network with DEGs was generated with the TCSBN database (http://inetmodels.com/) [19] through the NetworkAnalyst web server (https://www.networkanalyst.ca/) [18]. Results indicated that the co-expression networks consisted of 743 nodes and 876 edges (Figure 6 and Table 2).

Furthermore, a functional enrichment analysis with KEGG pathways revealed that the co-expression networks with DEGs were enriched in pathways related to tight junction, PPAR signaling pathway, mTOR signaling pathway, influenza A, tuberculosis, N-Glycan biosynthesis, terpenoid backbone biosynthesis, Notch signaling pathway, regulation of cyclin-dependent protein kinase activity, and steroid biosynthesis (Table 2). The GO BP term enrichment analysis showed those genes mostly involved in the establishment or maintenance of cell polarity, triglyceride metabolic process, protein targeting to membrane, defense response to virus, tuberculosis, post-translational protein modification, coenzyme biosynthetic process, gamete generation, transcription, DNA-dependent, positive regulation of translation, endoplasmic reticulum unfolded protein response, regulation of cyclin-dependent protein kinase activity, steroid biosynthetic process, regulation of the transcription of DNA-dependent, intra-Golgi vesicle-mediated transport term, and other rigid molecular machines in the biological process.

Based on the novel differentially regulated sub-networks detection tool, PhenomeScape [20], which could combine the fold changes of genes into the knowledge of networks and disease phenotypes, a series of differentially regulated sub-networks associated with phenotypes were identified with the random walk algorithm. In this research, seven phenotypes related to breast cancer were selected as the seed phenotypes (Table 5); subsequently, a total of 19 differentially regulated sub-networks enriched in the breast cancer phenotype related subnetwork were identified (Table 3). The sub-networks distributed by bufadienolide-like chemicals included RNA splicing (*p*-value = 2.00 × 10^−3^), apoptotic process (*p*-value = 2.00 × 10^−3^), extracellular matrix organization (*p*-value = 1.00 × 10^−3^), canonical Wnt signaling pathway (*p*-value = 2.20 × 10^−2^), synaptic transmission (*p*-value = 1.40 × 10^−2^), negative regulation of the JAK-STAT cascade (*p*-value = 4.20 × 10^−2^), adherens junction organization (*p*-value = 3.80 × 10^−2^), BMP signaling pathway (*p*-value = 4.10 × 10^−2^), negative regulation of cell migration (*p*-value = 1.30 × 10^−2^), and activation of signaling protein activity involved in the unfolded protein response (*p*-value = 1.90 × 10^−2^) (Figure 7).

The subnetwork A (Figure 7A), related to the RNA splicing function, was the first identified dysregulation subnetwork. It showed the genes involved in the mRNA splicing spliceosome were down-regulated, including the serine- and arginine- rich splicing factor members, SRSF4, SRSF5, and SRSF6, and peroxisome proliferator activated receptor gamma coactivator (PPARGC1A). The apoptotic process (Figure 7B) could have been dysregulated by bufadienolide-like chemicals, and this dysregulation was performed with the increased expression of SYT11, PARK2, PYHIN1, APC, RNF40, SERPINB3, TIAM2, ITSN1, SH3GL2, CASP1, GATA4, ITSN2, and PDE4DIP. Several cancer signaling pathways, including the Wnt signaling pathway, the JAK-STAT signaling pathway, and the BMP signaling pathway, also could had been dysregulated by bufadienolide-like chemicals (Figure 7D,F,H). This suggests that bufadienolide-like chemicals could increase the apoptotic process through a series of pathways or regulation networks. The subnetwork C (Figure 7C) was mostly related to the extracellular matrix organization being upregulated, including the genes, TIMP4, MMP3, SPARC, DPT, and ACAN. Also in this subnetwork, those genes that referred to the regulation of cell migration were downregulated, including the genes, TNFAIP6, DCN, SPARC, THBS1, and CCL8. This means the increase of the extracellular matrix may have hindered the migration of the tumor. Also, negative synaptic transmission, adherens junction organization, and regulation of cell migration was found in subnetwork E, G, and I (Figure 7E,G,I). Several metabolic processes were also discovered, including the drug metabolic process, xenobiotic metabolic process, oligosaccharide metabolic process, etc. All other PhenomeScape networks can be found in Appendix A.

Although, there is no evidence to prove the bufadienolide-like chemicals having obvious toxicity with the CEBS database (https://manticore.niehs.nih.gov/cebssearch/) [30]. In this research, in order to identify the potential cardiotoxicity of bufadienolide-like chemicals, 11 cardiotoxicity relation phenotypes (Table 6), including arrhythmia (HP:0011675), atrial fibrillation (HP:0005110), atrial flutter (HP:0004749), and other phenotypes, were chosen as seed phenotypes of cardiotoxicity with the aim of searching for the potential dysregulation subnetworks with cardiotoxicity. Results indicated six subnetwork related to membrane depolarization during the action potential (*p*-value = 3.70 × 10^−3^, Figure 8A), retinoic acid receptor binding (*p*-value = 2.00 × 10^−3^, Figure 8B), GABA receptor binding ((*p*-value = 3.00 × 10^−3^, Figure 8C), positive regulation of nuclear division (*p*-value = 5.00 × 10^−3^, Figure 8D), negative regulation of viral genome replication (*p*-value = 3.00 × 10^−3^, Figure 8E), and negative regulation of viral life cycle (*p*-value = 1.00 × 10^−3^), which were identified as potential cardiotoxicity subnetworks disturbed by bufadienolide-like chemicals (Table 4 and Figure 8). The subnetwork related to membrane depolarization may be the key potential cardiotoxic target of bufadienolide-like chemicals. These were also be observed by several widely used anticancer drugs with cardiotoxicity. For example, Adriamycin, Gleevec, and Herceptin were observed with a membrane depolarization appearance during clinical research [31,32].

Hub genes, mostly the highly connected nodes in the network, were identified by node degree and the MCC (Maximal clique centrality) algorithm with the Cytoscape plugin, cytoHubba [33]. Based on the threshold of the degree (degree > 5) and the MCC algorithm, 10 genes with MCC scores ranging from 126 to 953 were identified as hub genes (Figure 9A,B). Ten hub genes, including three 2′-5′-oligoadenylate synthetase genes, OAS1, OAS2, and OAS3; five interferon-induced genes, ISG15, IFIT1, IFI6, IFI44, and IFIL44L; and two other genes, including the kelch-like family member 35 (KLHL35) and Golgi Membrane Protein 1 (GOLM1) were identified. These were selected as the hub genes. Further investigation with TCGA [21] and the Kaplan-Meier databases [22] indicated that the 10 hub genes except KLHL35 were increased both in the treatment with bufadienolide-like chemicals and the TCGA breast cancer sample (Figure 9C). Six hub genes, including IFIT1, ISG15, IFI6, GOLM5, KLHL35, and OAS2, were associated with the total survival probability in breast cancer patients (Figure 9D). Further analysis of the correlation between the hub genes and the total survival time in breast cancer indicated that the high expression of GOLM5, KLHL35, and OAS2 was associated with a better survival probability.

## 3. Discussion

Recently, gene expression profile technology, including the microarray and RNA-seq, has been widely used to detect the potential mechanism of chemicals, however, a central problem still perplexes researchers on pharmacology and biology; that is, how chemicals disturb pathways and phenotypes through genes and their co-expression networks. In this research, with the use of bioinformatics tools, especially the differentially regulated sub-networks detection tools, PhenomeScape [20], CTD (http://ctdbase.org/) [16], and CMAP2 (https://portals.broadinstitute.org/cmap/) [17,18] databases, several dysregulated sub-networks related to the potential anticancer mechanism and cardiotoxicity were revealed, which was also further verified by the expression correlation and survival probability correlation with other databases. These results may highlight the potential molecular mechanism and application of bufadienolide-like chemicals on cancer, especially as a novel agent for breast cancer.

First, during the process of differentially expressed gene identification, in contrast to using the conventional method of differentially expressed gene selection with significance in statistics, a non-parametric unsupervised method of gene set variation analysis was used for differentially expressed gene identification. The results indicated a total of 80 DEGs involved in the 44 MSigDB C2 curated gene sets were identified (Figure 3A,B). After further analysis with the enrichment of the GO and KEGG pathway, we found genes that were up-regulated were most rich in their interferon signaling response to virus, defense to other organisms, regulation of viral genome replication, and 2′-5′-oligoadenylate synthetase activity. KEGG pathway enrichment analysis showed those genes could activate the IFN-induced pathway, type II interferon signaling pathway, and regulate the protein ISGylation pathway. However, the genes that were down-regulated were rich in protein kinase complex, transcription factor TFTC complex-1, SAGA- complex, and cargo loading into vesicle. KEGG pathway enrichment analysis showed those genes may be involved in negative transport of fringe-modified NOTCH to the plasma membrane pathway. By comparing the DEGs identification method with the statistical significance strategy, the number of DEGs enriched in MSigDB C2 curated gene sets may be much less compared to those DEGS with enrichment in the same biology function or similar pathway. Also, the same results were proven by the examples of the GSVA package [15].

Second, during the process of similar small molecule detection, CTD (http://ctdbase.org/) [16] and CMAP2 (http://www.broadinstitute.org/cMAP/) [17,18] databases were used. The results indicated that the bufadienolide-like chemicals had the same effect as valproic acid and estradiol. Valproic acid is a histone deacetylase inhibitor, and it was shown to inhibit proliferation via Wnt/β catenin signaling activation. Estradiol was also proven to have anticancer activity, especially in postmenopausal women. Also, the evidence from the CTD database (http://ctdbase.org/) indicated bufadienolide-like chemicals have the potential ability to be used as hormones and anticancer and vasoprotectives agents.

Third, during the process of co-expression network reconstruction and dysregulated sub-networks detection, a novel plug of PhenomeScape was used, which could combine the data of gene expression into the knowledge of protein–protein interaction networks and disease phenotype [20]. During the analysis with the damaged osteoarthritic cartilage gene expression profile, several significant sub-networks related to damaged osteoarthritic cartilage were identified: Mitotic cell cycle, Wnt signaling, apoptosis, and matrix organisation [34,35]. In this research, with the PhenomeScape tool [20], a total of 19 differentially regulated sub-networks were identified, and 10 sub-networks were proven to relate to breast cancer by evidence, including RNA splicing, apoptotic process, cell migration, extracellular matrix organization, adherens junction organization, synaptic transmission, and so on. Also, with the PhenomeScape tool [20], six dysregulated subnetworks, including the subnetwork related to membrane depolarization during the action potential, retinoic acid receptor binding, GABA receptor binding, positive regulation of nuclear division, negative regulation of viral genome replication, and negative regulation of viral life cycle, were identified. Those dysregulated subnetworks may play important roles in the cardiotoxicity of bufadienolide-like chemicals.

Hub gene selection and its relation to survival probability indicated that 10 hub genes (except KLHL35) were increased in both breast cancer and samples treated with bufadienolide-like chemicals. Further analysis in relation to the total survival probability showed six hub genes, including IFIT1, ISG15, IFI6, GOLM5, KLHL35, and OAS2, were associated the total survival time and high expression of GOLM5, KLHL35, and OAS2 was associated with better survival probability.

## 4. Materials and Methods

### 4.1. Microarray Data Information

The gene expression profiles of GSE85871 (https://www.ncbi.nlm.nih.gov/gds/), which is a gene expression profile treated with 102 chemicals from Chinese traditional medicine, and is based on the Affymetrix GPL571 platform (Affymetrix Human Genome U133A 2.0 Array, Santa Clara, CA, USA), was submitted by Lv et al. [36].

In this study, the raw data of 4 controls and 14 samples treated with bufadienolide-like chemicals (1 μM and treatment with 12 h), including resibufogenin, bufalin, arenobufagin, cinobufagin, bufotoxin, telocinobufagin, bufotaline, and cinobufotali, were downloaded from the GEO database via GEOquery [37] packages in the R3.5.1 [38] environment.

### 4.2. Identification of DEGs Associated with Relative Enrichment Pathways

In order to obtain a series of differentially expressed genes (DEGs) with biological interpretation, a novel R package, GSVA [15], was employed, which allowed the assessment of the DEGs underlying pathway activity variation by transforming the gene expression profile into the prior knowledge of the gene set. In accordance with MIAME (Minimum Information About a Microarray Experiment) standards [39,40], the DEGs disturbed by bufadienolide-like chemicals were identified by a series of standard flow with the R environment. First, the quality assessments, background correction, and normalization were preprocessed and normalized with the affy [41] and gcrma [42] packages. Then, the batch effects were examined and removed with the combat and sva functions in the SVA (Surrogate Variable Analysis) package [43]. Subsequently, a non-specific probes filtering step was performed with the nsFilter function in the genefilter package [44], the quality control probes of Affymetrix, probe sets without Entrez ID annotation, probesets whose associated Entrez ID was duplicated in the annotation, and the top 20% with smaller variability were first removed. Finally, the GSVA [43], GSEABase [45], limma [46] package, and c2BroadSets from Molecular Signatures Database (MSigDB) [47,48] were used to select the DEGs enriched in the relative enrichment pathways.

During the process of DEGs selection with relative enrichment sets, the gene expression profile was first transformed into the prior knowledge gene sets of c2BroadSets and the enriched gene sets were selected with the screening criteria of FDR < 0.01. Then, the DEGs enriched in the c2BroadSets gene sets were selected with the limma [46] package, and the screening criteria were set with FDR < 0.01 and |logFC| > 1. The DEGs associated with relative enrichment pathways were used for further analysis.

During the process of DEGs identification, the Biobase [49] package and GSVAdata [50] package were also applied. The results were visualized with the ggplot2 [51], ggpubr [52], pheatmap [53], and cowplot [54] packages.

### 4.3. Gene Enrichment Analysis

In order to obtain a comprehensive understanding of those genes involved in the prior knowledge of gene sets, GO and KEGG enrichment analysis were performed with the clueGO plug [23] in Cystoscape [24]. The significantly enriched GO terms and KEGG pathways were calculated by the hypergeometric test [55], and cut-off criteria were set as FDR < 0.05. Another statistical parameter of the Kappa Score were set as middle stringency, which means the terms in the network were merged with the middle related terms based on their overlapping genes. The minimum percentage and minimum genes enriched in GO terms or KEGG pathways were set as 1.0% and 2; also, the term fusion parameter was also chosen. Other options, including the statistical options, reference options, grouping options, and visual options, were set with the default setting.

### 4.4. Similar Small Molecule Detection

In order to detect the similar small molecules with bufadienolide-like chemicals, the DEGs with up or down were respectively submitted to the CTD (http://ctdbase.org/) [16] and CMAP2 (http://www.broadinstitute.org/cMAP/) database [16,17]. During the process of detection of similar small molecules with the CTD database, the threshold of degree in the degree filter network was set as 10. During the process of detection of similar small molecules with the connectivity map database, the enrichment score and *p*-value were chosen as the similarity index between the gene expression profile of the query signature and that of chemicals in the CMAP2 database.

Also, the potential toxicity the same as bufadienolide-like chemicals were also detected with the CEBS database (https://manticore.niehs.nih.gov/cebssearch/) [30], but there was no evidence to prove the bufadienolide-like chemicals had obvious toxicity.

### 4.5. Gene Co-Expression Network Analysis and Disease Phenotype Association

To obtain a comprehensive understanding of the potential mechanism of DEGs involved in breast cancer, co-expression network analysis, phenome association, and survival correlation analysis were investigated with the NetworkAnalyst database (https://www.networkanalyst.ca/) [56] and PhenomeScape plug [20] in Cystoscape [24]. Also, other plugs and databases, including the cytoHubba [33], TCSBN database (http://inetmodels.com/) [19], TCGA database [21] and Kaplan-Meier (KM) plotter database (http://kmplot.com/) [22], and the Phenomiser (http://compbio.charite.de/phenomizer/) [57] web tool, were also used for hub gene selection and survival correlation analysis. First, the breast mammary tissue-specific co-expression networks were investigated with the TCSBN database (http://inetmodels.com/) through the NetworkAnalyst web server (https://www.networkanalyst.ca/). The GO and KEGG enrichment terms of networks were also investigated with the NetworkAnalyst web server (https://www.networkanalyst.ca/). Subsequently, the differentially regulated sub-networks enriched in genes associated with the breast cancer phenotype were identified by random sampling (10,000 sub-networks) methods with the PhenomeScape plug and Phenomiser (http://compbio.charite.de/phenomizer/) web tool. First, through the search with Phenomiser web tool and the manual of UberPheno ontology [57], 6 breast carcionma phenotypes (Table 5) and 11 cardiotoxicity relation phenotypes (Table 6) were chosen as the potential anticancer mechanism or potential cardiotoxicity association phenotype. Parameters of the maximum initial sub-network size of 7 and an empirical *p*-value threshold of 0.05 were used for filtering the differentially regulated sub-networks enriched in genes associated with breast cancer or the cardiotoxicity phenotype.

Hub genes, highly interconnected with nodes in the network, are considered functionally significant in the network. In our study, the top 10 hub genes were defined by the node degree and MCC algorithm in the Cytoscape plugin, cytoHubba [33]. We used the previously described workflow that selected the essential proteins from the yeast protein interaction network with the MCC algorithm [33]. First, the degrees of nodes were computed by the NetworkAnalyzer [58] in Cytoscape. Then, the nodes with a degree greater than a threshold were selected as potential candidate hub genes, and the threshold was the maximum integer as $2\times \underset{v\in V,\;Deg\left(v\right)>t}{\sum }Deg\left(v\right)>\underset{v\in V,}{\sum }Deg\left(v\right)$, where v is the collection of nodes within the network V, *Deg*(*v*) is the degree of node v. Last, the top 10 hub genes were ranked by the MCC algorithm in the cytoHubba plugin. The hub genes common in breast tissue co-expression networks were chosen as the candidates for further validation with TCGA [21] and the Kaplan-Meier (KM) plotter database (http://kmplot.com/analysis/) [22].

## 5. Conclusions

In this research, with a serious of bioinformatics analysis, we noticed that the bufadienolide-like chemicals may perform anticancer activity through RNA splicing, apoptotic process, cell migration, extracellular matrix organization, adherens junction organization, synaptic transmission, Wnt signaling, AK-STAT signaling, BMP signaling pathway, and the unfolded protein response (Figure 10A). Also, further investigation of the potential cardiotoxicity of bufadienolide-like chemicals indicated the dysregulated subnetwork related to membrane depolarization during the action potential, retinoic acid receptor binding, GABA receptor binding, positive regulation of nuclear division, negative regulation of viral genome replication, and negative regulation of viral life cycle may play important roles in cardiotoxicity (Figure 10B). Additionally, those may highlight the potential molecular mechanism of bufadienolide-like chemicals on breast cancer, but still, there are several problems with no better solution, including the renal toxicity of bufadienolide-like chemicals, and the difference of potential molecular mechanisms among different stem nuclei in bufadienolide-like chemicals was also clearly illuminated in this research.

## Figures and Tables

**Figure 1 cancers-11-00091-f001:**
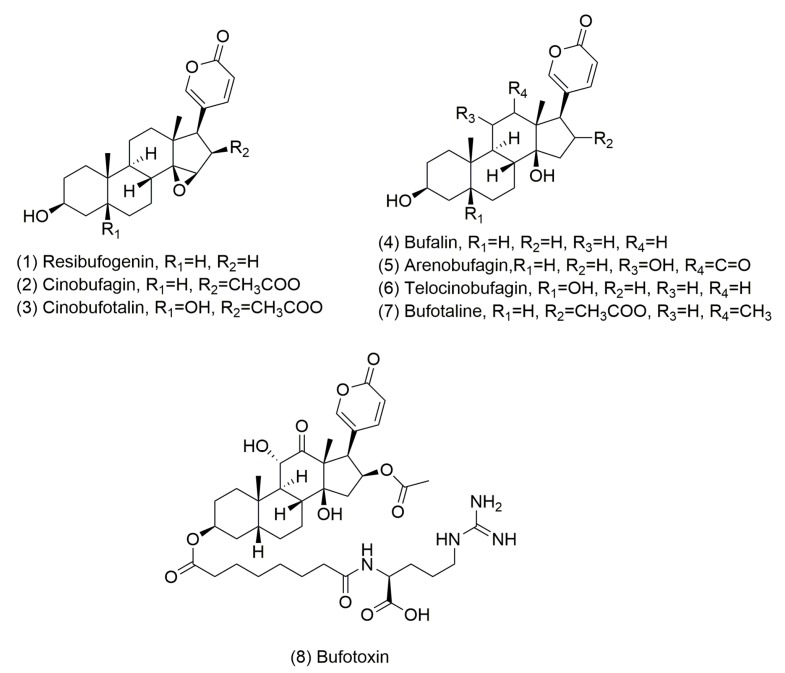
The structural formula of the eight bufadienolide-like chemicals.

**Figure 2 cancers-11-00091-f002:**
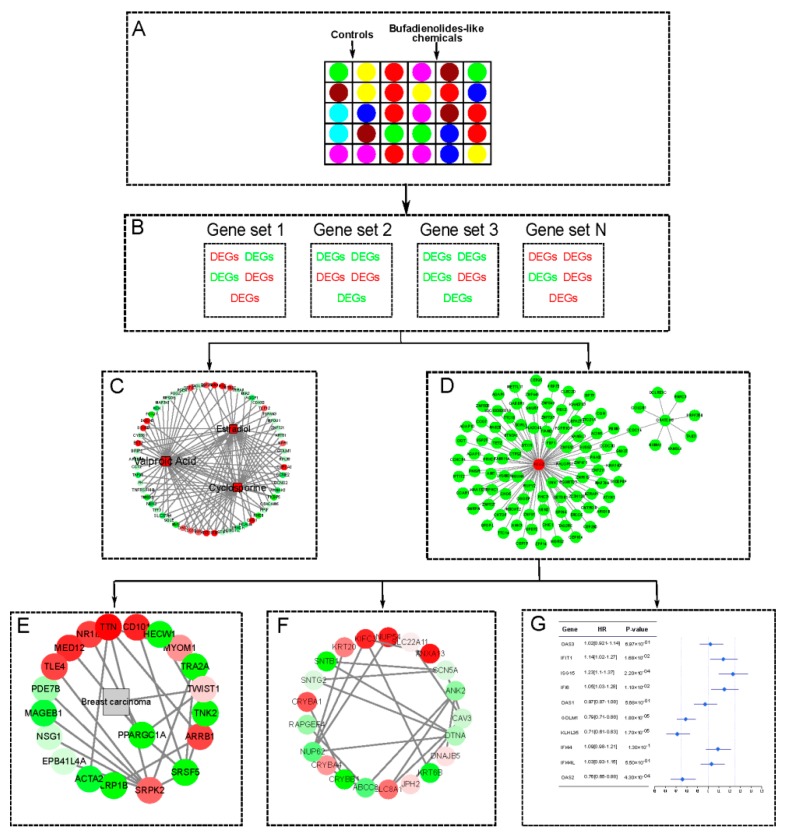
The workflow to study the potential mechanism of bufadienolides-like chemicals on breast cancer via bioinformatics analysis. (**A**) The experimental design and basic information of this analysis. (**B**) The DEGs’(Differentially expressed genes) identification with the GSVA (Gene set variation analysis) algorithm [15]. (**C**) Similar small-molecule detection with the Comparative Toxicogenomics Database (CTD) [16] and connectivity map (CMAP2) [17,18] database. (**D**) The tissue-specific co-expression network constructed with the TCSBN (Tissue and cancer specific biological networks) database [19]. (**E**) The differentially expressed subnetworks detected with the UberPheno database and PhenomeScape plug [20]. (**F**) The arrhythmia-related subnetworks detected with the UberPheno database and PhenomeScape plug [20]. (**G**) The expression and survival relation of hub genes validated by TCGA (The Cancer Genome Atlas) [21] and the Kaplan-Meier (KM) plotter databases [22].

**Figure 3 cancers-11-00091-f003:**
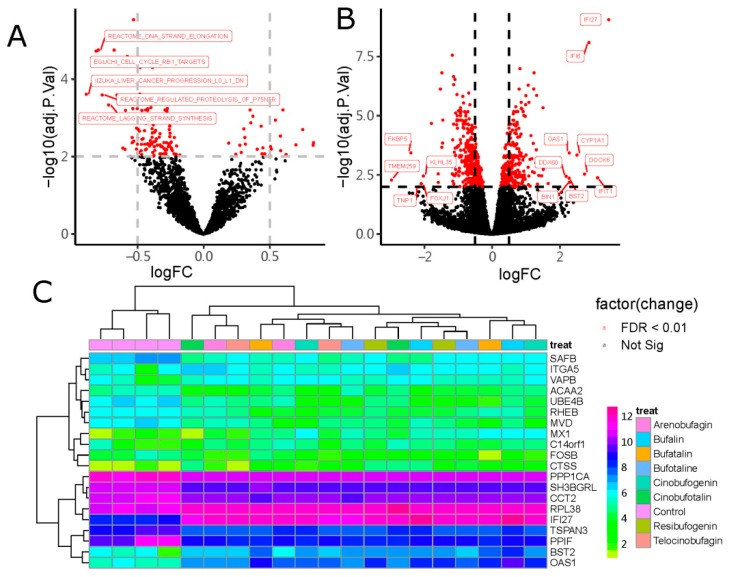
The differentially expressed genes (DEGs) disturbed by bufadienolide-like chemicals through the gene set variation analysis (GSVA) algorithm. (**A**) The differentially expressed gene sets disturbed by bufadienolide-like chemicals (|logFC| ≥ log2(2) and adjPvalue < 0.001). (**B**) The DEGs relate to differentially expressed gene sets (|logFC| ≥ log2(2) and adjPvalue < 0.001). (**C**) The heatmap of the top 20 DEGs disturbed by bufadienolide-like chemicals.

**Figure 4 cancers-11-00091-f004:**
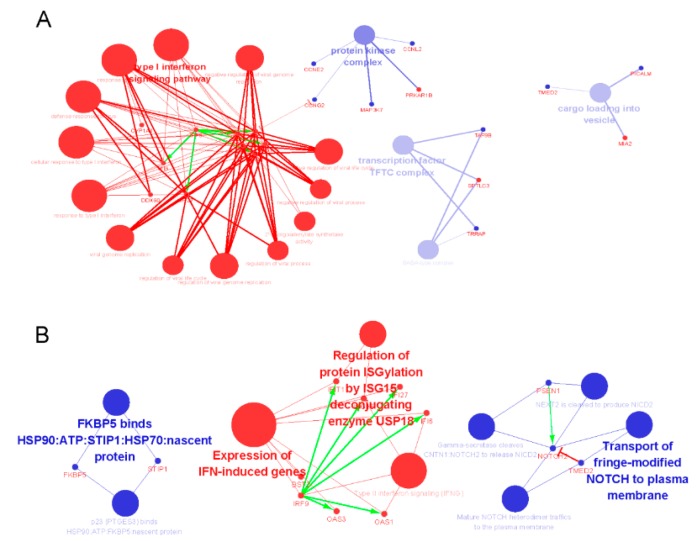
The GO and KEGG enrichment result of DEGs disturbed by bufadienolide-like chemicals. (**A**) Representative biomolecular network of GO enrichment terms. The bigger red nodes imply enrichment of GO terms with up-regulated genes. The bigger blue nodes suggest enrichment of GO terms with down-regulated genes. The small red nodes imply up-regulated genes. The small blue nodes are down-regulated genes. Undirected edges imply enrichment, green directed edges are activated according to the string database. The red directed edges implies suppression from the evidence generated by the String database. (**B**) Representative biomolecular network of KEGG enrichment term, the nodes, and edges also had the same means with Figure 4A.

**Figure 5 cancers-11-00091-f005:**
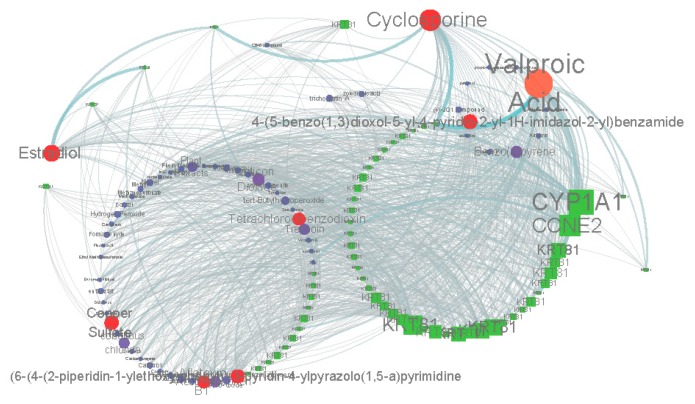
Chemicals-gene interaction network for the DEGs disturbed by bufadienolide-like chemicals. Square nodes represent the DEGs. Circle nodes represent the chemicals predicted by the CTD Database. The size of the nodes represents the degree. Circle nodes with red represent the similar small molecule predicted by degree (degree ≥ 30).

**Figure 6 cancers-11-00091-f006:**
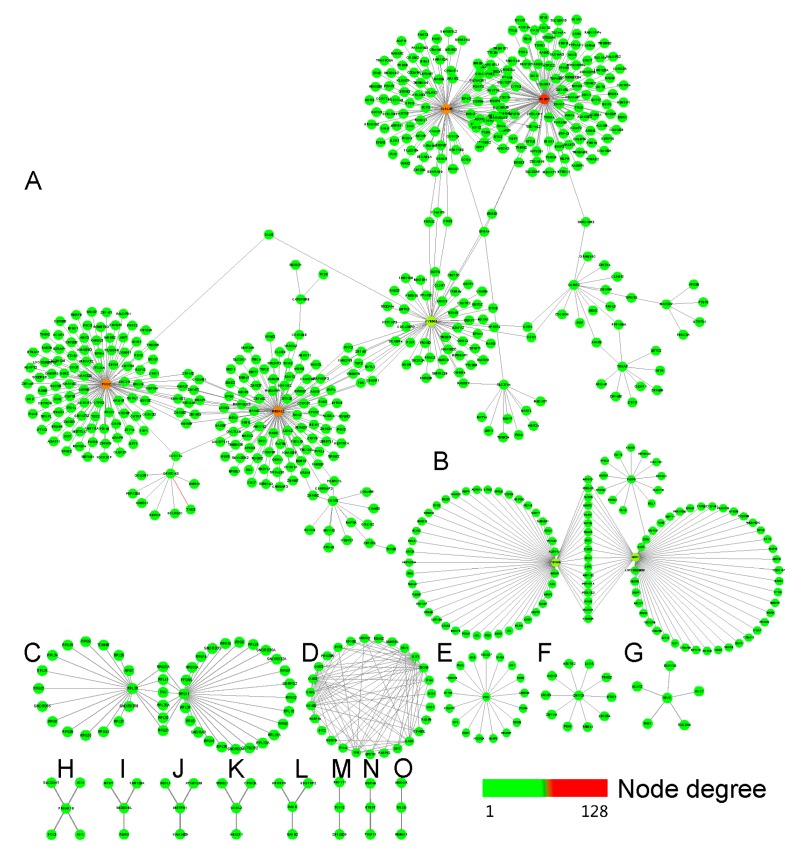
The breast tissue specific co-expression network with DEGs generated by the TCSBN (Tissue and cancer specific biological networks) database (http://inetmodels.com/) through the NetworkAnalyst (https://www.networkanalyst.ca/) web server. (**A**–**O**), the subnetworks of co-expression network origin from the seeds of DEGs.

**Figure 7 cancers-11-00091-f007:**
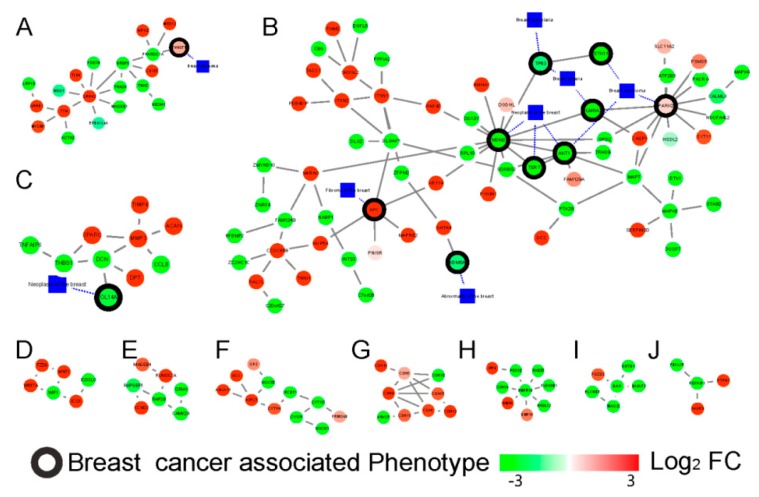
The differentially expressed networks regulated by bufadienolide-like chemicals, and generated by the PhenomeScape plug. Sub-networks linked to breast cancer, RNA splicing (2.00 × 10^−3^) (**A**), apoptotic process (2.00 × 10^−3^) (**B**), extracellular matrix organization (1.00 × 10^−3^) (**C**), canonical Wnt signaling pathway (2.20 × 10^−2^) (**D**), synaptic transmission (1.40 × 10^−2^) (**E**), negative regulation of JAK-STAT (Janus kinase/signal transducers and activators of transcription) cascade (4.20 × 10^−2^) (**F**), adherens junction organization (3.80 × 10^−2^) (**G**), BMP signaling pathway (4.10 × 10^−2^) (**H**), negative regulation of cell migration (1.30 × 10^−2^) (**I**), and activation of signaling protein activity involved in the unfolded protein response (1.90 × 10^−2^) (**J**). The fold change of the proteins is shown by the node color, and breast cancer-associated phenotype annotated proteins were used to generate the sub-networks and are shown with a black border.

**Figure 8 cancers-11-00091-f008:**
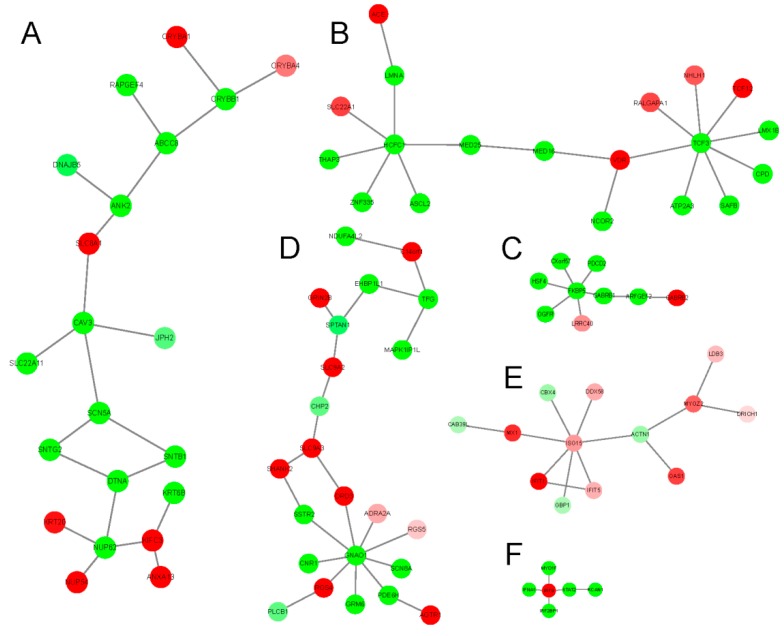
The differentially regulated sub-networks with potential cardiotoxicity disturbed by bufadienolide-like chemicals, generated by the PhenomeScape plug with seeds of 11 cardiotoxicity phenotypes. (**A**) Subnetwork related to membrane depolarization during action potential (3.70 × 10^−2^), (**B**) Subnetwork related to retinoic acid receptor binding (2.00 × 10^−3^), (**C**) Subnetwork related to GABA receptor binding (3.00 × 10^−3^), (**D**) Subnetwork related to positive regulation of nuclear division (5.00 × 10^−3^), (E) subnetwork related to negative regulation of viral genome replication (3.00 × 10^−3^), and (**F**) subnetwork related to negative regulation of viral life cycle (1.00 × 10^−3^).

**Figure 9 cancers-11-00091-f009:**
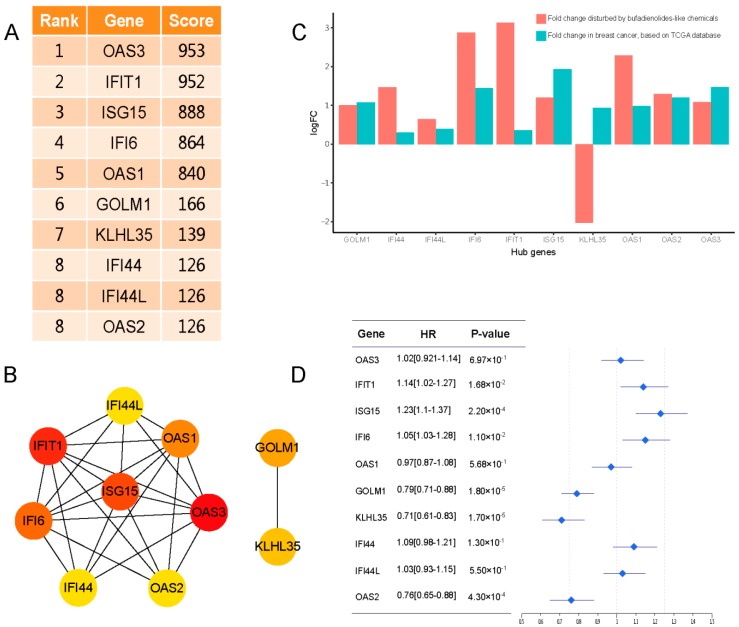
The 10 hub genes and their correlation with the total survival probability in breast cancer. (**A**) The 10 hub genes and their MCC (Maximal clique centrality) score. (**B**) The network of hub genes. (**C**) The expression correlation with breast cancer, validated by the TCGA database. (**D**) The total survival probability correlation with breast cancer, validated by the Kaplan-Meier (KM) plotter database.

**Figure 10 cancers-11-00091-f010:**
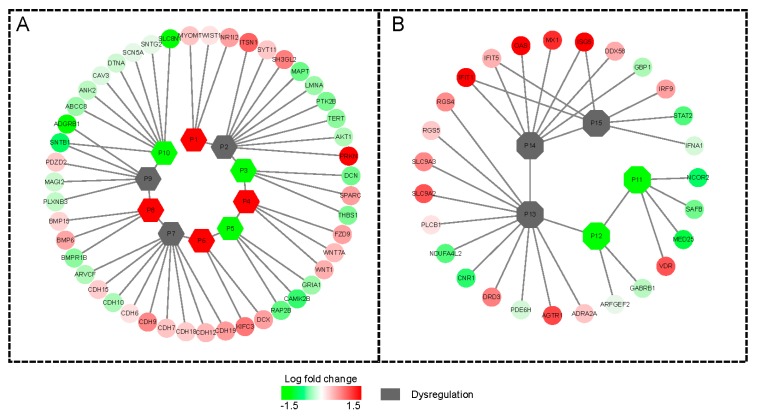
The potential anticancer mechanism and cardiotoxicity of bufadienolide-like chemicals. (**A**) The potential anticancer mechanism of bufadienolide-like chemicals. Nodes p1–p10 means the 10 differentially regulated sub-networks in Figure 7. (**B**) The potential cardiotoxicity of bufadienolide-like chemicals: Node p11–p15 means the five differentially regulated sub-networks in Figure 8.

**Table 1 cancers-11-00091-t001:** Top 20 CMAP2 (connectivity map, https://portals.broadinstitute.org/cmap/) hits correlated with bufadienolide-like chemicals’ treatment.

Rank	ATC Code	Mean Score	Enrichment	*p*-Value	Specificity
1	V03AF	−0.471	−0.71	4.45 × 10^−3^	3.82 × 10^−2^
2	G03GB	0.449	0.655	3.29 × 10^−2^	7.47 × 10^−2^
3	C05AX	0.41	0.689	1.95 × 10^−2^	4.76 × 10^−2^
4	C05CX	0.41	0.689	1.95 × 10^−2^	4.76 × 10^−2^
5	D07XC	−0.372	−0.661	1.44 × 10^−3^	8.10 × 10^−3^
6	N05BE	−0.359	−0.719	1.26 × 10^−2^	1.22 × 10^−2^
7	C08EA	0.292	0.539	1.87 × 10^−2^	1.45 × 10^−1^
8	N05AC	0.259	0.365	2.32 × 10^−3^	3.90 × 10^−1^
9	D06BB	−0.252	−0.405	9.39 × 10^−3^	1.44 × 10^−1^
10	D06BX	−0.249	−0.72	3.74 × 10^−3^	1.38 × 10^−2^
11	N02BB	0.244	0.404	2.71 × 10^−3^	1.75 × 10^−2^
12	N02CX	0.189	0.481	3.16 × 10^−2^	4.43 × 10^−2^
13	A07EA	−0.186	−0.343	6.96 × 10^−3^	2.55 × 10^−2^
14	S02BA	−0.167	−0.383	5.03 × 10^−3^	1.31 × 10^−2^
15	B01AC	0.152	0.243	2.71 × 10^−2^	1.19 × 10^−1^
16	S03BA	−0.144	−0.366	2.02 × 10^−2^	4.80 × 10^−2^
17	R03BA	−0.141	−0.29	1.19 × 10^−2^	4.00 × 10^−2^
18	S01CB	−0.136	−0.326	1.21 × 10^−2^	2.61 × 10^−2^
19	R01AD	−0.113	−0.266	4.30 × 10^−3^	4.83 × 10^−2^
20	C07AA	−0.109	−0.262	1.14 × 10^−2^	2.22 × 10^−1^

**Table 2 cancers-11-00091-t002:** The tissue specific co-expression network regulated by bufadienolide-like chemicals and their enrichment with GO and KEGG.

Subnetwork Number	Nodes	Edges	Seeds	KEGG Enrichment		GO Enrichment	
				KEGG Pathway	*p*-Value	BP Term	*p*-Value
A	492	558	13	Tight junction	4.19 × 10^−4^	Establishment or maintenance of cell polarity	2.83 × 10^−4^
B	113	128	3	PPAR signaling pathway	7.75 × 10^−6^	Triglyceride metabolic process	1.25 × 10^−7^
C	46	50	2	mTOR signaling pathway	9.62 × 10^−3^	Protein targeting to membrane	4.93 × 10^−67^
D	27	86	6	Influenza A	3.04 × 10^−10^	Defense response to virus	1.24 × 10^−22^
E	18	17	1	Tuberculosis	2.01 × 10^−4^	Tuberculosis	2.01 × 10^−4^
F	11	10	1	N-Glycan biosynthesis	9.19 × 10^−3^	Post-translational protein modification	6.33 × 10^−3^
G	6	5	1	Terpenoid backbone biosynthesis	1.72 × 10^−4^	Coenzyme biosynthetic process	1.55 × 10^−5^
H	5	4	1	Notch signaling pathway	2.98 × 10^−2^	Gamete generation	1.34 × 10^−2^
I	4	3	1	Null	Null	Transcription, DNA-dependent	1.31 × 10^−2^
J	4	3	1	Null	Null	Positive regulation of translation	1.17 × 10^−2^
K	4	3	1	Null	Null	Endoplasmic reticulum unfolded protein response	6.51 × 10^−3^
L	4	3	1	Regulation of cyclin-dependent protein kinase activity	1.24 × 10^−2^	Regulation of cyclin-dependent protein kinase activity	1.24 × 10^−2^
M	3	2	1	Steroid biosynthesis	7.68 × 10^−3^	Steroid biosynthetic process	2.07 × 10^−6^
N	3	2	1	Null	Null	Regulation of transcription, DNA-dependent	1.84 × 10^−2^
O	3	2	1	Null	Null	Intra-Golgi vesicle-mediated transport	4.47 × 10^−3^

**Table 3 cancers-11-00091-t003:** Summary of differentially regulated sub-networks disturbed by bufadienolide-like chemicals.

Subnetwork Number	No. of Nodes	GO-BP	Empirical *p*-Value
**A**	21	RNA splicing	2.00 × 10^−3^
**B**	73	apoptotic process	2.00 × 10^−3^
**C**	11	extracellular matrix organization	1.00 × 10^−3^
**D**	6	canonical Wnt signaling pathway	2.20 × 10^−2^
**E**	7	synaptic transmission	1.40 × 10^−2^
**F**	11	negative regulation of JAK-STAT cascade	4.20 × 10^−2^
**G**	9	adherens junction organization	3.80 × 10^−2^
**H**	9	BMP signaling pathway	4.10 × 10^−2^
**I**	6	negative regulation of cell migration	1.30 × 10^−2^
**J**	4	activation of signaling protein activity involved in unfolded protein response	1.90 × 10^−2^
**K**	12	drug metabolic process	1.20 × 10^−2^
**L**	6	negative regulation of lipid storage	4.50 × 10^−2^
**M**	6	xenobiotic metabolic process	1.70 × 10^−2^
**N**	8	relaxation of cardiac muscle	4.80 × 10^−2^
**O**	5	very long-chain fatty acid metabolic process	1.70 × 10^−2^
**P**	4	oligosaccharide metabolic process	3.10 × 10^−2^
**Q**	4	collagen catabolic process	2.50 × 10^−2^
**R**	4	response to cocaine	2.70 × 10^−2^
**S**	4	behavioral response to nicotine	4.20 × 10^−2^

**Table 4 cancers-11-00091-t004:** Summary of differentially regulated sub-networks with potential cardiotoxicity disturbed by bufadienolide-like chemicals.

Subnetwork Number	No. of Nodes	GO-BP	Empirical *p*-Value
**A**	21	Membrane depolarization during action potential	3.70 × 10^−2^
**B**	19	Retinoic acid receptor binding	2.00 × 10^−3^
**C**	9	GABA receptor binding	3.00 × 10^−3^
**D**	23	Positive regulation of nuclear division	5.00 × 10^−3^
**E**	13	Negative regulation of viral genome replication	3.00 × 10^−3^
**F**	6	Negative regulation of viral life cycle	1.00 × 10^−3^

**Table 5 cancers-11-00091-t005:** UberPheno phenotype terms selected for identification of the differentially regulated sub-network with the potential anticancer mechanism of bufadienolide-like chemicals.

Phenotype ID	Phenotype Description
HP:0100783	Breast aplasia
HP:0100013	Neoplasm of the breast
HP:0003002	Breast carcionma
HP:0003187	Breast hypoplasia
HP:0000769	Abnormality of the breast
HP:0010619	Fibroma of the breast

**Table 6 cancers-11-00091-t006:** UberPheno phenotype terms selected for identification of the differentially regulated sub-network with potential cardiotoxicity of bufadienolide-like chemicals.

Phenotype ID	Phenotype Description
HP:0011675	Arrhythmia
HP:0005110	Atrial fibrillation
HP:0004749	atrial flutter
HP:0011215	Hemihypsarrhythmia
HP:0002521	Hypsarrhythmia
HP:0040182	Inappropriate sinus tachycardia
HP:0001962	Palpitations
HP:0005115	Supraventricular arrhythmia
HP:0004755	Surpraventricular tachycardia
HP:0004308	Ventricular arrhythmia
HP:0011841	Ventricular flutter

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
