# Peer review of "The Potential Mechanism of Bufadienolide-Like Chemicals on Breast Cancer via Bioinformatics Analysis"

_cancers, 2019, doi:10.3390/cancers11010091_

Reviewer 1 Report

The manuscript entitled “Explore the potential mechanism of bufadienolides-like chemicals on breast cancer through Bioinformatics analysis” investigated the potential mechanism of bufadienolides-like chemicals against breast cancer using a series of bioinformatics analysis. This study provided useful information for understanding the role of bufadienolides-like chemicals on breast cancer, which provided an effective basic for clinic use of bufadienolides-like chemicals.

Overall, this study has significant reference value to the application of bioinformatics in cancers, and its innovation is also worthy of recognition compared to the existing reports. But a few points in manuscript need more explanations before publication:

1. The toxicity of bufadienolides-like chemicals, which been only investigated with CEBS database, is not enough, and still need to be further investigated with other databases and software.

2. The authors also performed an association analysis between the differentially regulated sub-networks and 7 disease phenotypes. However, the authors did not mention what are these 7 disease phenotypes? They should explain it in the manuscript. Moreover, they should specifically point out that which sub-networks are significantly associated with each disease phenotype, respectively.

3. In the section of “abstract”, the results of this research hadn’t been clearly described, and a detailed description is required.

4. In the section of “discussion”, a diagram with summary of potential mechanism maybe needed

5. In the section of “materials and methods”, also a diagram maybe helpful to understand the workflow of this work

6. In the part of “2.1. Identification of DEGs”, the DEGs in gene sets should be compared with the reference gene sets, please check those information carefully.

7. In the part of “2.1. Identification of DEGs”, the annotation of Figure 1A, 1B, 1C should be revised as Figure 2A, 2B, 2C

8. In the part of “2.1. Identification of DEGs”, the DEGs involved in the critical pathways indicated by GO or KEGG analysis should be demonstrated in detail. I suggest the authors to draw a heatmap for the crucial genes and systematically demonstrate the possible function of candidate genes.

9. In section 2.2 (Line 123), the authors mentioned that they performed a tissue specific co-expression network analysis. However, through the whole manuscript, the authors did not illustrate how many tissues (and how many samples in each tissue) in total were used. I strongly recommend the authors to explain this in the METHOD section.

10. In the part of “2.2. The tissue specific co-expression network and breast cancer associated subnetwork regulated by bufadienolides-like chemicals”, Line 150-151: 'a total of 23 differentially regulated sub-networks enriched in breast cancer phenotype related subnetwork were identified (Table 2)'. But, there were actually only 19 records in Table 2. The number of sub-networks should be checked and make corresponding modification.

11. In the part of “4.1 Microarray data information”, the concentrations of chemicals, treat time and replicates should be described.

12. In the part of “4.2 Identification of DEGs associated with relative enrichment pathways”, the standard for identification of DEGs should be described in detail, especially for the comparing method between control and treated samples.

13. In the part of “4.4 Gene co-expression network analysis and disease phenotype association”, the effective link related to database and software should be supplied, please add those information.

14. In the part of “4.4 Gene co-expression network analysis and disease phenotype association”, the TCGA [23] database is inconsistent with the reference “Kosinski M, B.P., RTCGA: The Cancer Genome Atlas Data Integration. R package version 1.12.0. 2018.” Please check it and make corresponding modification.

15. The grammar of the manuscript need to be revised carefully.

16. Some typing errors in the manuscript need to be revised, such as "Breast cancer" in line 34, “Chinese Traditional medicine" in line 38; the line spacing of Line 47 to 69, 225-228, 316-325, and the text alignment of line 35-44, 73-77, 114-122, 143-145, 178-185, 199-202, 225-228.

17. The format of the reference list needs to be checked again.

Author Response

Dear professor,

       Thank you very much for the suggest on this manuscript, and I had checked the mistake in this manuscript,  also the grammar had been revised by Letpub Co.Ltd.

1. The toxicity of bufadienolides-like chemicals, which been only investigated with CEBS database, is not enough, and still need to be further investigated with other databases and software.

Checked and replenished the potential cardiotoxicity analysis.

2. The authors also performed an association analysis between the differentially regulated sub-networks and 7 disease phenotypes. However, the authors did not mention what are these 7 disease phenotypes? They should explain it in the manuscript. Moreover, they should specifically point out that which sub-networks are significantly associated with each disease phenotype, respectively.

Checked and demonstrated in line 191-196, also in section of “4.5 Gene co-expression network analysis and disease phenotype association” (Line 392-397). And the results of relation between sub-networks to disease phenotype could be found in Fig 7.

3. In the section of “abstract”, the results of this research hadn’t been clearly described, and a detailed description is required.

Checked and rearranged the section of “abstract”.

4. In the section of “discussion”, a diagram with summary of potential mechanism maybe needed

Checked and demonstrated in Fig10.

5. In the section of “materials and methods”, also a diagram maybe helpful to understand the workflow of this work

Checked and demonstrated in Fig2.

6. In the part of “2.1. Identification of DEGs”, the DEGs in gene sets should be compared with the reference gene sets, please check those information carefully.

Checked and demonstrated in line 93~112

7. In the part of “2.1. Identification of DEGs”, the annotation of Figure 1A, 1B, 1C should be revised as Figure 2A, 2B, 2C

Checked

8. In the part of “2.1. Identification of DEGs”, the DEGs involved in the critical pathways indicated by GO or KEGG analysis should be demonstrated in detail. I suggest the authors to draw a heatmap for the crucial genes and systematically demonstrate the possible function of candidate genes.

Checked, and demonstrated in Fig4., and “the small red nodes imply up-regulated genes. the small blue nodes are down-regulated genes. ”

9. In section 2.2 (Line 123), the authors mentioned that they performed a tissue specific co-expression network analysis. However, through the whole manuscript, the authors did not illustrate how many tissues (and how many samples in each tissue) in total were used. I strongly recommend the authors to explain this in the METHOD section.

Checked and this analysis were performed with breast cell line MCF7  

10. In the part of “2.2. The tissue specific co-expression network and breast cancer associated subnetwork regulated by bufadienolides-like chemicals”, Line 150-151: 'a total of 23 differentially regulated sub-networks enriched in breast cancer phenotype related subnetwork were identified (Table 2)'. But, there were actually only 19 records in Table 2. The number of sub-networks should be checked and make corresponding modification.

Checked

11. In the part of “4.1 Microarray data information”, the concentrations of chemicals, treat time and replicates should be described.

Checked

12. In the part of “4.2 Identification of DEGs associated with relative enrichment pathways”, the standard for identification of DEGs should be described in detail, especially for the comparing method between control and treated samples.

Checked and in line 349-354

13. In the part of “4.4 Gene co-expression network analysis and disease phenotype association”, the effective link related to database and software should be supplied, please add those information.

Checked and demonstrated in Fig7.

14. In the part of “4.4 Gene co-expression network analysis and disease phenotype association”, the TCGA [23] database is inconsistent with the reference “Kosinski M, B.P., RTCGA: The Cancer Genome Atlas Data Integration. R package version 1.12.0. 2018.” Please check it and make corresponding modification.

Checked and RTCGA is a R package Integrated the TCGA information

15. The grammar of the manuscript need to be revised carefully.

Checked and had been modified by LetPub Co.Ltd

16. Some typing errors in the manuscript need to be revised, such as "Breast cancer" in line 34, “Chinese Traditional medicine" in line 38; the line spacing of Line 47 to 69, 225-228, 316-325, and the text alignment of line 35-44, 73-77, 114-122, 143-145, 178-185, 199-202, 225-228.

Checked and had been modified

17. The format of the reference list needs to be checked again.

Checked and had been modified

Reviewer 2 Report

This is an interesting study that applied comprehensive bioinformatics studies to explore the potential mechanisms bufadienolides-like chemicals on breast cancer. Gene set variation analysis, tissue specific co-expression network sub-networks detection with disease phenome, hub gene selection and gene signatures of bufadienolides-like chemicals were used in the experimental design. Results suggested that  bufadienolides-like chemicals is involved in the signaling of RNA splicing, apoptotic process, cell migration, extracellular matrix organization, and Wnt signaling, AK-STAT signaling. The data implied the anti-cancer effects of bufadienolides-like chemicals. Basically, this is a well-designed bioinformatics study that may provide a novel way for drug discovery. Application this platform is able to speed up the pharmacological identification of new compounds. The only concern in this manuscript is the functional study. None of the functional studies was attached in this study. Without the biological/physiological validation, the influence of  bioinformatics analysis designed in this study is limited.  

Author Response

Dear professor,

       Thank you very much for the suggest on this manuscript, and I had checked the mistake in this manuscript,  also the grammar had been revised by Letpub Co.Ltd.

This is an interesting study that applied comprehensive bioinformatics studies to explore the potential mechanisms bufadienolides-like chemicals on breast cancer. Gene set variation analysis, tissue specific co-expression network sub-networks detection with disease phenome, hub gene selection and gene signatures of bufadienolides-like chemicals were used in the experimental design. Results suggested that bufadienolides-like chemicals is involved in the signaling of RNA splicing, apoptotic process, cell migration, extracellular matrix organization, and Wnt signaling, AK-STAT signaling. The data implied the anti-cancer effects of bufadienolides-like chemicals. Basically, this is a well-designed bioinformatics study that may provide a novel way for drug discovery. Application this platform is able to speed up the pharmacological identification of new compounds. The only concern in this manuscript is the functional study. None of the functional studies was attached in this study. Without the biological/physiological validation, the influence of  bioinformatics analysis designed in this study is limited.  

Reply: Checked, but unfortunately, our team couldn’t replenish the biological/physiological validation experiment in short time. And the main idea of this work is illuminate the potential mechanism with systematics, especially the dysregulation subnetwork related to breast cancer phenome. If add the biological/physiological validation experiment, the balance of the article or the meaning may had changed.  Besides, the expression or  survival  relation of hub genes had been validated with TCGA data.

Reviewer 3 Report

Authors in this manuscript used published another study (14 MCF7 samples treated with bufadienolides-like chemicals vs 4 control by microarray) to perform more and different analysis. Instead of using traditional microarray differential expression analysis methods, they performed Gene Set Variation Analysis to identify differentially expressed genes, which were then used as seed input to the network analysis. 

They concluded that bufadienolides-like chemicals had the most same target with valproic and estrdiol and had the potential ability to be used as anticancer agents.

The results are interesting, but the greatest weakness with this manuscript is the lack of further validation of the conclusion using independent data set or different experiments (rt-pcr or west blot, etc). 

The manuscript is also littered with grammatical and typographical errors. The authors need to get editing help from someone with full professional proficiency in English.

Author Response

Dear professor,

 Thank you very much for the suggest on this manuscript, and I had checked the mistake in this manuscript, also the grammar had been revised by Letpub Co.Ltd.

(1)The results are interesting, but the greatest weakness with this manuscript is the lack of further validation of the conclusion using independent data set or different experiments (rt-pcr or west blot, etc). 

Reply: Checked, but unfortunately, our team couldn’t replenish the biological/physiological validation experiment in short time. And the main idea of this work is illuminate the potential mechanism with systematics, especially the dysregulation subnetwork related to breast cancer phenome. If add the biological/physiological validation experiment, the balance of the article or the meaning may had changed.  Besides, the expression or  survival  relation of hub genes had been validated with TCGA data.

(2)The manuscript is also littered with grammatical and typographical errors. The authors need to get editing help from someone with full professional proficiency in English.

Reply: Checked and had been modified by LetPub Co.Ltd

Round  2

Reviewer 3 Report

Previous comments have been addressed.